# New Insights into Osteointegration and Delamination from a Multidisciplinary Investigation of a Failed Hydroxyapatite-Coated Hip Joint Replacement

**DOI:** 10.3390/ma13214713

**Published:** 2020-10-22

**Authors:** Florian Schönweger, Christoph M. Sprecher, Stefan Milz, Corina Dommann-Scherrer, Christoph Meier, Alex Dommann, Antonia Neels, Peter Wahl

**Affiliations:** 1Division of Orthopaedics and Traumatology, Cantonal Hospital Winterthur, 8400 Winterthur, Switzerland; christoph.meier@ksw.ch (C.M.); peter.wahl@ksw.ch (P.W.); 2Division of Orthopaedics and Traumatology, Regional Hospital Lugano, 6900 Lugano, Switzerland; 3Biomedical Materials Group, AO Research Institute Davos, 7270 Davos, Switzerland; christoph.sprecher@aofoundation.org; 4Department of Neuroanatomy, Ludwig Maximilian University, 80336 Munich, Germany; stefan.milz@med.uni-muenchen.de; 5Institute of Pathology, Cantonal Hospital Winterthur, 8400 Winterthur, Switzerland; corina.dommann@ksw.ch; 6Centre for X-ray Analytics, Empa, Swiss Federal Laboratories for Material Science and Technology, 8600 Dübendorf, Switzerland; Alex.Dommann@empa.ch (A.D.); antonia.neels@empa.ch (A.N.); 7ARTORG Centre for Biomedical Engineering Research, University of Berne, 3012 Berne, Switzerland

**Keywords:** hydroxyapatite, coating, delamination, arthroplasty, histology, XRD

## Abstract

Hydroxyapatite (HA) coatings have become very popular in uncemented total hip arthroplasty (THA). Analysis of retrievals and tissue samples from an HA-coated femoral stem, which failed within 14 months after THA, provides exceptional insights into the failure mechanism, as well as the process of osteointegration of such an implant. Methods: Retrievals were photo-documented. Samples were examined by micro-computed tomography, X-ray diffraction (XRD) and embedded in polymethylmethacrylate for histology. Results: The coating had partially delaminated. The sandblasted surface of the stem was partially polished by the delaminated HA coating, indicating failure before revision. In the tissue samples, the HA coating was well integrated by newly formed bone trabeculae. No adverse biological reaction was observed. XRD analysis showed that residues of the HA coating were still present and could clearly be differentiated from the surrounding bone. Preferential orientation of the HA crystallites could be identified within the newly formed bone, representing a potential mechanical weakness induced either by physiologic strain or by the coating. Conclusion: current HA coatings, relatively thick and made of high crystallinity HA, may be prone to delamination, as also seen in our study. Recent efforts have aimed towards thinner (<1 μm) coatings with nanocrystalline HA structures that possibly relate to lower delamination risks. However, the question arises if HA coatings are beneficial since sandblasted non-coated stems offer similar results without the risk of delamination. XRD not only permits differentiation between the HA from the coating and the HA of the ongrown bone, it also provides new insights into the microstructure of this newly formed bone.

## 1. Introduction

Uncemented fixation is being used more and more frequently in hip arthroplasty [1,2]. To achieve the long-term stability of uncemented implants, either ingrowth or ongrowth of bone on the surface of the implant is necessary [3]. A certain porosity of the implant surface is needed for this to happen, which depends on the materials used and surface properties [3,4,5,6,7]. Porosity is obtained either by grit blasting the surface of the metal alloy or through addition of a coating [3,6]. Most commonly, coatings either consisting of titanium or hydroxyapatite (HA) are applied by plasma-spray [3,6].

Animal and clinical studies have shown excellent ongrowth of bone onto HA coatings [5,7,8,9,10,11,12,13,14,15]. Even if questionable, HA coatings are believed to potentially accelerate osteointegration [5,7,12,14,15,16]. However, adding a coating might also be disadvantageous, as any material interface represents a potential mechanical weakness, with consecutively possible delamination [16], particularly as the modulus of elasticity of HA and titanium alloys differ, and as eccentric loads femoral stems that are exposed to total hip arthroplasty (THA) lead to shear forces at the interface [17].

In this report, we present results from an advanced and multidisciplinary analysis of a single clinical case of early failure of an uncemented, HA-coated total hip arthroplasty (THA). The study questions were as follows. Did the HA coating delaminate? Did bone grow onto the HA coating? Is it possible to differentiate between the HA from the coating and the HA originating from the ongrown bone by x-ray diffraction (XRD) methods? Despite an analysis limited to a single retrieval, new insights into the failure mechanism and osteointegration were gained. The methodology of the described combined analytical tools may serve as an algorithm to investigate samples and retrievals from other revisions of orthopaedic implants, particularly as appropriate analytical methods are not commonly available in clinical routines.

## 2. Patients and Methods

### 2.1. Case Description

A 68-year-old deaf male patient (60 kg, 175 cm, BMI 19.6 kg/m^2^) with clinically manifest osteoporosis suffered a femoral neck fracture following a low-energy fall on his right hip. The initial treatment with uncemented THA was performed elsewhere (Versafit CC Trio cup with highly-crosslinked polyethylene liner, 32 mm CoCr head, Quadra-H stem; Medacta, Castel San Pietro, Switzerland), through a direct anterior approach on a traction hemi-table (Figure 1). The stem was undersized by two sizes compared to preoperative templating and implanted in varus position in relation to the axis of the femoral diaphysis (Figure 1A). The patient recovered from the operation without complications and was able to return to his partially institutionalised everyday life. A postoperative leg-length discrepancy, however, hampered ambulation, with recurrent tripping.

Approximately 14 months postoperatively, following another low-energy fall on his right hip, the patient suffered a multifragmentary periprosthetic fracture of the proximal femur with loosening of the stem (Figure 1B). Corresponding to a periprosthetic fracture unified classification system (UCS) type B2 [18], stem revision was indicated. Surgery was performed through a transfemoral approach to the hip, formally completing exposure through the fracture fragments [19,20]. This approach ensures good distal purchase of the uncemented, tapered, fluted, modular revision stem (Revitan straight, Zimmer Biomet, Winterthur, Switzerland) used [21]. The stem in situ was loose and could be retrieved without any instrumentation. Delamination of the HA coating from the stem had already been suspected on the preoperative conventional radiographies (Figure 1C). Fragments from the bone at the implant interface could be chiselled off easily from the endostal surface of the medial cortex of the femur for further analysis before reaming for the new stem. The tissue samples were fixated and sent for further examination in a buffered formaldehyde solution (4%) (Formafix, Hittnau, Switzerland). Good osteointegration of the cup was confirmed, and it was therefore left in place, despite excessive anteversion of 28°, and measured in the plane of the lateral inclination [22,23]. However, to reduce the risk of dislocation, the liner was exchanged to accommodate a 36 mm head [22]. Internal fixation of the proximal femur was achieved with cerclage cables (Dall-Miles 2.0 mm; Stryker, Kalamazoo, MI, USA). As the femoral offset could not be reconstructed anatomically with the new stem, a certain degree of leg-lengthening had to be accepted, to avoid instability of the hip.

Microbiological samples and routine histopathological analysis showed no signs of infection. Postoperative recovery was uneventful. Follow-up more than two years postoperatively was uneventful, except that the leg length discrepancy had to be compensated with shoe sole raise on the contralateral side.

### 2.2. X-ray Micro Computed Tomography (µCT)

The sampled tissue fragments were examined by computed tomography (µCT40; Scanco, Bassersdorf, Switzerland), with an isotropic voxel size of 10 µm. For 3D reconstruction, bone was identified as voxels with a density ranging 250–1200 mg HA/ccm. Tissue with a higher density was considered to correspond to residues of the HA coating.

### 2.3. Histology

The histological processing of the sampled tissue contained a dehydration step with ascending series of ethanol (70, 80, 96, and 100%, with two exchanges for each step, CLN GmbH, Freising Germany), followed by a transfer to xylene (VWR, Darmstadt, Germany), and finally to methyl-methacrylate (Merck, Darmstadt, Germany) for embedding. Serial sections with an initial thickness of 200 µm were obtained using a saw-microtome (Leica SP 1600; Leica, Wetzlar, Germany). The sections were then glued on plastic slides (Acrylics, Vink Kunststoffe, Glching, Germany), ground using grinding papers with different grain sizes (Exakt Micro Grinding System; Exakt Apparatebau, Norderstedt, Germany), polished, and surface stained with a 15% Giemsa and 1% Eosin solution (Fluka; Sigma-Aldrich Chemie, Buchs, Switzerland). The final thickness of the stained sections was 120 ± 20 µm. Histological sections were microscopically analyzed on an Axiophot microscope (Carl Zeiss, Oberkochen, Germany) with transmitted illumination and imaged with a color image analysis system (Axiocam HRc, AxioVision V 4.8.2., Carl Zeiss, Oberkochen, Germany).

### 2.4. X-ray Diffraction (XRD)

The interface between the implant and the HA coating, respectively, between the ongrown bone and remnants of the HA coating was investigated using x-ray diffraction methods. Crystallographic phase analysis was performed on the residues of the HA coating found on the retrieved stem and compared with the coating of a new stem of the same model. Additionally, the crystallinity and the respective HA orientation profiles were analyzed.

Samples of bone and residues of the HA coating taken from the implant surface were measured on an Imaging Plate Diffractometer System (IPDS-II) from STOE and Cie (Darmstadt, Germany) equipped with a graphite monochromator. Data collection was performed at room temperature using Mo-Kα radiation (λ = 0.71073 Å, beam diameter 0.5 mm). Two-dimensional diffraction images (30 min per exposure) were obtained at an image plate distance of 200 mm. Intensity integration was performed over the entire image (360°).

The cross section of the interface between the implant coating and ongrown bone was investigated using a Bruker D8 Discover Davinci diffractometer (Bruker AXS, Karlsruhe, Germany) equipped with a Göbel mirror for the generation of a parallel beam and with a LynxEye 1D detector using Cu-Kα radiation (λ = 1.5406 Å, line beam width = 1 mm). 2θ/ω scans were carried out between 20° and 100° in 2θ with overlapping 1 mm steps, with the linear x-ray beam moving from the depth of the implant to the coating, respectively, the bone on the surface. For phase analysis, the software Diffrac.Eva V4.1 (Bruker, Karlsruhe, Germany) was used in combination with the crystal open database COD [24].

## 3. Results

### 3.1. Surface Observations on the Retrieval

Macroscopic examination of the stem revealed various processes with heterogeneous distribution (Figure 2). Bone had ongrown in small patches on the stem, particularly anteriorly and posteriorly. Nearly no residues of the HA coating could be identified. Zones of partial polishing of the grit-blasted surface of the metal alloy were visible. Optical microscopy of the patches of ongrown bone revealed brittle residues of HA coating left only in these areas, the bone having a cancellous structure (Figure 3). Light reflections on the metal surface hindered obtaining useful microphotographs of the polished areas.

### 3.2. µCT

The µCT reconstructions showed close contact of the radiologically dense layer, corresponding to the HA coating, with bone trabeculae (Figure 4). All coating fragments were oriented in one plane and were 60–100 µm thick. Few dispersed HA particles <20 µm in size were scattered within the tissues adjacent to the layer of coating. No more HA coatings could be identified, within the bone trabeculae or in the depth of the tissue sample.

### 3.3. Histology

PMMA-embedded histological sections of the undecalcified tissue samples showed an intimate contact of HA coating residues with trabecular bone (Figure 5). Osteocytes present in nearly all osteocyte lacunae confirmed bone vitality. Bone trabeculae showed various stages of maturation, including woven bone, indicating ongrowth of newly formed bone structures. As is characteristic for an elderly patient, the bone marrow contained predominantly fat cells and additionally few isolated dark grayish particles <20 µm in size localized near the HA coating, corresponding visually to residues of the coating. Focal fibrosis and vascular proliferation were also observed. No histiocytic, neutrophilic, or lymphoplasmocytic inflammatory infiltrates were present, and only sporadic multinucleated giant cells could be detected. No residues of hemosiderin (i.e., previous hematoma) were identified.

### 3.4. XRD

While the surface of a new implant was entirely covered with a white powder-like structure showing larger crystallites, the surface of the retrieved stem showed only residues of white powder-like substance, mostly overlaid by a darker, partially transparent trabecular structure, corresponding to cancellous bone (Figure 3). XRD analysis (Figure 6) confirmed the white crystalline powder as being pure but partially degraded HA, and the darker, more transparent structure as bone ongrown on the implant (Figure 6A). The x-ray pattern obtained from a scratched-off sample of the coating from a new stem showed sharp diffraction peaks, indicating the high crystallinity of the HA, with crystallites > 100 nm (Figure 6B). A low background signal indicated that no additional amorphous material was present. The HA diffraction patterns of the pristine coating and of the brittle residues of white powder on the explanted stem were nearly identical, indicating that the initial coating was still partially present in between and most probably also under the ongrown trabecular bone. The structure of the HA of the ongrown bone trabeculae showed much broader reflections than the HA from the coatings, which is related to its nano-crystalline character. Additionally, the presence of protein residues such as collagen was confirmed by the high background signal and the broad peak between 5 and 10° in 2Theta (Mo-radiation), as well as from the high scattering intensity in the small angle X-ray scattering (SAXS) region (Figure 6A).

Diffraction patterns illustrated in Figure 7 show the successive transition from the phase of the TiAlNb alloy of the stem (hexagonal, P6_3_/mmc with a = 2.95 and c = 4.69Å) to the highly crystalline HA coating (HA-coating, hexagonal, P6_3_/m with a = 9.42 and c = 6.89Å) and finally to the HA of the ongrown bone (HA-bone). While the phase of the HA-coating is identifiable by sharp reflections within the diffraction profile, the HA from the ongrown bone shows wider reflections related to the nano crystalline character of the hydroxyapatite. The crystallite size can be calculated using the Scherrer equation (LIT). The (002) reflection of the hydroxyapatite is used, along with the LaB_6_ for the correction of the instrumental broadening. The crystallite size determined for the growing (preferential) direction is 23 nm, and is a bit smaller than the one for human bone-derived HA with 31 nm (LIT). The appearance of a higher background signal and additional broad peaks at lower angles is related to the organic material within the osteoid, such as collagen fibers.

One-dimensional diffraction patterns for all three samples, the powder obtained from the coating of a new implant, as well as samples obtained from the two distinguishable structures on the retrieved stem, the coating residues, and the trabecular bone, as illustrated in Figure 3, were superposed in Figure 8.

A piece of trabecular bone sampled from the surface of the retrieved stem was analyzed at various angles, without rotation during measurement. Different sectors of the diffraction image were integrated and resulted in one-dimensional diffraction patterns showing different intensity distributions (Figure 9). Preferred orientation was found for the (001) crystallographic direction (see the change in intensities of the reflections (002) and (004) in Figure 9), which can directly be related to directional or preferential orientation of the HA within the newly formed bone.

## 4. Discussion

We had an exceptionally rare occasion to retrieve and analyze materials and tissues from a patient requiring early revision of his THA. This case provides insights into a failure mechanism easily overlooked, as well as new insights into the process of osteointegration of uncemented implants and of the biodegradation of HA coatings.

Proper photodocumentation of the retrieval is a simple and very useful tool. As illustrated in Figure 2, various elements of osteointegration and failure could be identified, and appear to have a heterogeneous distribution along the surface of the implant. Bone had grown onto most of the surface, particularly anteriorly and posteriorly. Virtually no more residues of HA coating were identifiable on the exposed areas without bone. Polishing of some areas of the grit-blasted surface could also be seen. This could not be analyzed further by optical microscopy due to light reflexes on the surface, a commonly known issue in retrieval analysis. As the stem was loose and could be retrieved without any instrumentation, no surgical artifacts interfered with interpretation. Chiseling off any ongrown bone would have left rough isolated marks on the surface of the metal, which were not present, except for the transverse osteotomy of the transfemoral approach (Figure 2) [19,20]. The initially roughened metallic surface of the titanium stem could have been polished only by repeated friction with a harder substance, which solely might be the HA from the coating or from the surrounding bone. This proves instability and fretting preceding the fracture, which required revision.

An osteoporotic bone in combination with an undersized stem, due to a subtle varus malalignment at the primary operation (Figure 1), may be seen as contributing factor to the fracture. Once loosened, the stem could act as a pestle, favoring the development of a periprosthetic fracture following a minor trauma. Uncemented THA is known to be associated with an increased risk of periprosthetic fracture, increasing steadily over the long term [25]. Undersizing of the stem might also be considered as a contributing factor to the fracture risk. Then, newly formed cancellous bone has to fill the space between the cortex and the stem, as observed in this case (Figure 1, Figure 2, Figure 3, Figure 4 and Figure 5), instead of direct contact with cortical bone and integration by the inner side of the bone from the cortex. Osteoporosis primarily affects the metabolically more active cancellous bone, thus creating a supplementary mechanical weakness [26]. A detailed analysis of all available radiographies often provides many useful insights in case of failure of orthopaedic implants.

Undersizing of the stem allowed development of trabecular bone between the cortex and the implant, which was then accessible to sampling. The measurements performed by µCT (Figure 4) showed a HA coating thickness between 60 and 100 µm. This corresponds to the 80 µm of coating thickness specified by the manufacturer of this particular stem model. The variability of our measurements may be explained by the variability of the coating thickness caused by the plasma spray application during manufacture [27], as well as by inaccuracies of the measurement on tissue biopsies by µCT. While only one image is shown (Figure 4), the same pattern was identifiable on all fragments available for analysis. Thus, it can be derived that no relevant degradation of the HA coating occurred in the areas where bone had ongrown (Figure 3 and Figure 4). However, the HA had dissolved between the bone trabeculae (Figure 3 and Figure 4). This irregular dissolution pattern of HA coatings is also apparent in retrieval studies, even if not explicitly described by the authors [10,11,13,15]. A direct ongrowth of bone on the metal surface of the implant might therefore be impeded by the HA coating. If bone ongrown on the HA coating hinders its dissolution, then the coating represents a foreseeable failure point, as delamination will invariably happen, particularly as the bond of HA with the underlying metal weakens during in vivo degradation of the coating [16].

In areas where bone had ongrown on the HA coating, newly formed and vital bone in good contact with the HA layer was identified in the histology sections (Figure 4). No areas of local necrosis or inflammatory reaction were detectable. This is consistent with previous studies indicating the osteoconductive effect of HA [8,9,10,11,13,15]. The bonding between HA and bone was solid, and a separation between the two components was not noticed. No direct ongrowth of bone trabeculae in areas with dissolved HA coating was observed. Fragmentation of HA coating was also visible on the surface of the retrieved implant (Figure 3). This has also been described in animal studies with longer periods of observation [14,27]. Mechanical properties of the HA coating are weakened by this fragmentation, favoring delamination [14,27]. Every interface is a mechanical weakness, particularly if the mechanical properties of the materials differ, as is the case with HA and titanium alloy. Thus, it is most likely that such an interface is prone to material failure. This is supported by an animal push-out model of ingrown implants in which delamination is the common failure mode of HA coatings [16]. However, delamination of the HA coating was discarded in a postmortem study, as it was thought it was caused during stem retrieval [10]. While being a well-known failure mode of any interfaces, to the best of our knowledge, delamination of HA coatings is not otherwise reported in arthroplasty. PMMA-embedded histology is a technically demanding technique, but allows interfaces between bone and implants to be preserved. Paraffin-embedded histology, the clinical standard procedure, suffers from cutting artefacts, and decalcification would simply dissolve HA or calcium phosphate coatings.

Analysis of the HA coating from a new stem with XRD showed a highly crystalline form of HA, with no additional amorphous phases being present at the interface between the metal surface and the coating. No alteration of the coating material occurred during manufacture. This was also true for the retrieved stem (Figure 7). When analysing the material chipped off the retrieved stem, additional smaller crystallites were visible, indicating degradation in vivo. Their derivation from the HA of the coating could be proven using XRD (Figure 8).

We were able to show a preferential orientation of the HA crystallites within the newly formed bone on the implants surface (Figure 9), indicating a directional growth pattern of the trabecular bone. This might be due to physiologic strains on the implant, but might also been induced by the coating. Preferential orientation of the crystalline structure of the ongrown bone, however, represents a supplementary mechanical weakness to shear forces. This represents an additional potential failure point and a potential explanation for the concomitant presence of ongrown bone as well as polished areas (Figure 2). Preferential orientation of the surrounding bone trabeculae along a well-integrated stem also had been noticed in another retrieval study, but was not interpreted in detail [28]. In this case, suboptimal orientation of an undersized stem may, however, have contributed to this feature.

HA may have osteoconductive effects, which is true for many other surfaces as well. Acceleration of bone ongrowth, an often-cited feature, is not a question of chemical properties but of topography, as HA coatings have a more pronounced porosity than common grit-blasted metals [5,7]. While contact surface for the bone might be increased by an HA coating, bone mineralization becomes weaker [9]. HA coatings may not only increase but also reduce primary stability of hip implants, with consecutively increased migration [29,30]. Increased porosity, however, reduces mechanical resistance of HA coatings [27]. In the end, no clinical advantage for HA-coated stems is identifiable in hip arthroplasty. When comparing single stem designs with and without HA coating, revision rates for all causes, as well as for aseptic loosening in particular, do not differ [31,32,33]. Although clear evidence of clinically relevant benefits of HA coatings is lacking, they are used more and more commonly in THA [1,2] despite increased costs in manufacturing and potential for specific complications associated with any coating. Although our case report describes a single event, the analyses carried out independently using various methods came to the same results, demonstrating delamination of the HA coating that must have preceded the fracture leading to the revision. While this is just a single case, failure analysis is a well-recognized way to improve medical care and technology, particularly for rare and undescribed events.

HA coatings may, however, be improved in the near future. Recent studies show that thin HA coatings (<1 μm) with nano-crystalline character reduce the risk of delamination and favor osteointegration [34]. By trying to use biologic-like materials, the trend is moving towards development of thin film applications for implants. Biomimetic calcium-phosphate coatings, which resemble natural bone apatite by introducing magnesium, carbonate, strontium and potassium, can boost bone regeneration to a higher extent than stoichiometric HA can [34]. In recent publications, the role of HA crystallite size and HA coating thickness in osteointegration have been discussed and evaluated by XRD [34,35]. In-vitro studies have shown that amorphous calcium phosphates can transform to nanostructured, platelet-like bone apatite crystals [36], resulting in a higher mechanical stability and a lower tendency to cracking and delamination [37], with the possibility to obtain a nanostructured surface texture able to boost platelet, protein, and cell adhesion [38,39,40]. However, there are still unclear stages in the process of bone mineralization and bone ongrowth on implants.

## 5. Conclusions

Delamination of the HA coating from the stem was not only proven, but must have preceded the fracture leading to revision. The coating was state-of-the-art, without contamination or phase alteration. In the present case, the coating supported initial osteointegration and formed a stable link with the surrounding bone. However, the mechanical link between the metallic implant and the coating did not resist over time. Failure of arthroplasty caused by delamination of the HA coating is most probably underreported due to the lack of adequate tissue sampling as well as due to the technical complexity and difficulties for proper workup. Although our report describes only a single case, the different analyses point towards the same conclusion. Refined analysis provides new insights into the process of osteointegration of HA-coated implants as well as the process of dissolution of the HA coating. Particularly, HA coatings might even impede ongrowth of bone on the implant surface. Preferential orientation of the HA crystallites within ongrown bone might be due to physiologic strain, but might also be induced by the coating, and this represents a supplementary mechanical weakness which may relate to the failure.

## Figures and Tables

**Figure 1 materials-13-04713-f001:**
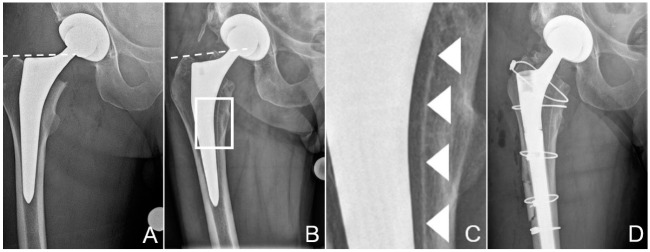
Zone of interest of the anteroposterior radiographies of the pelvis after total hip replacement (**A**) and after the patient had suffered the periprosthetic fracture of the proximal femur (**B**). Dashed lines at the tip of the greater trochanter help identify posttraumatic sintering of the stem. Note that the stem was undersized, with more than 1 mm between the stem and the inner cortex. Scale-up of (**B**), marked by a white rectangle, is provided in (**C**) to better illustrate the dense line, most likely the hydroxyapatite (HA) coating, separated from the stem, marked by white arrowheads. These tissues could be sampled for further analysis. (**D**) shows the condition after revision. The osteotomy, which was necessary to retrieve the stem, is visible, as are the cerclage cables used for fixation of the fracture and the osteotomy. Additionally, note the thinner liner in the cup, to accommodate the larger head.

**Figure 2 materials-13-04713-f002:**
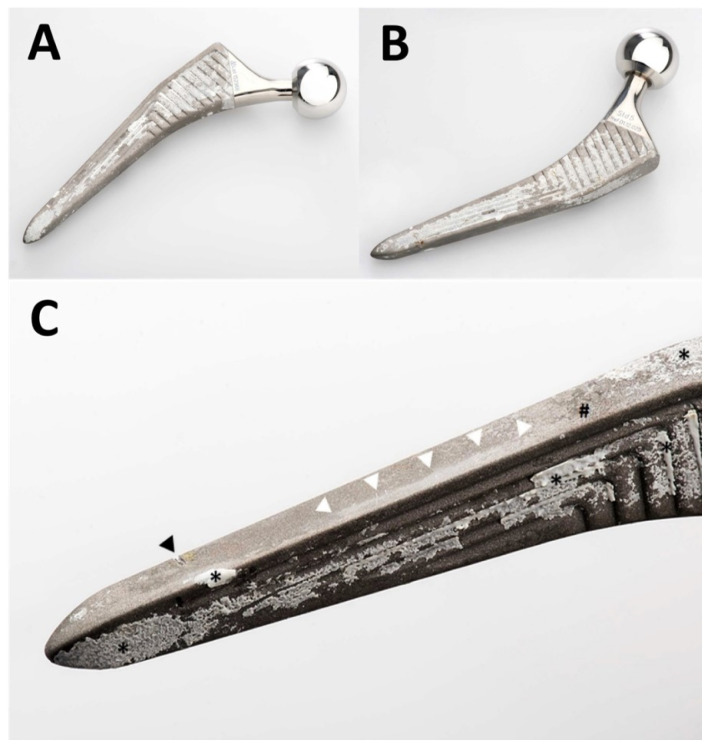
Photographs of the retrieved Quadra-H stem. An overview of the anterior aspect is shown in (**A**) and of the posterior aspect in (**B**). A greater magnification of the anterolateral aspect of the distal part of the stem is shown in (**C**). The femur had already fractured, and the stem could be retrieved without any instrumentation. The transverse nick near the tip (black arrowhead in (**C**)) was caused by the transverse osteotomy of the transfemoral approach. This was the only damage caused by instruments. Note unevenly distributed patches of ongrown bone on the stem’s surface (* in (**C**)). Only small patches of residues of the HA coating (# in (**C**)) are visible. The near edge of the stem presents an extensive area of polishing (white arrowheads in (**C**)) of the grit-blasted surface of the metal alloy.

**Figure 3 materials-13-04713-f003:**
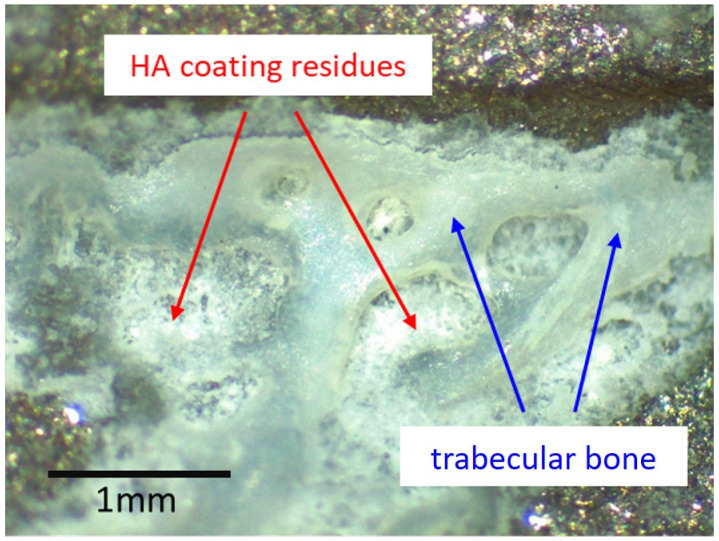
Optical microscopy image of one of the patches of ongrown bone at the surface of the retrieved stem. Light reflexes from the metal surface limit such analysis to areas with limited light reflection. No HA residues could be identified where no bone was present. Where trabecular bone was present, brittle residues of the HA coating could be identified. The same pattern was observable wherever ongrown bone trabeculae were present.

**Figure 4 materials-13-04713-f004:**
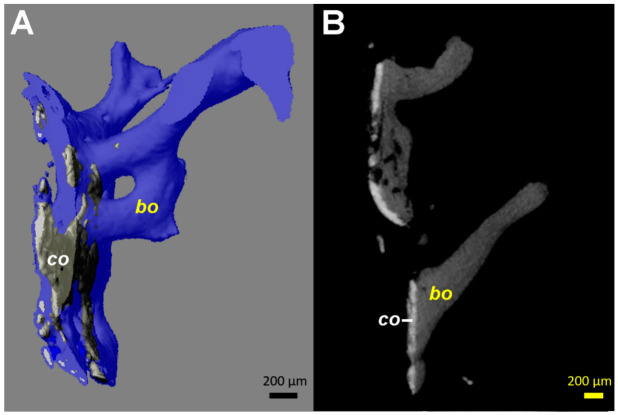
Computed tomography (µCT) images from the tissue samples collected at the interface between the stem and the medial cortical bone (area marked by arrowheads in Figure 1C). Identical patterns could be identified on all fragments available, even if the images from only one are illustrated. (**A**) 3D reconstruction in semi-transparent mode shows the close contact of the coating material (co, colored in grey) with bone (colored in blue). The coating ca has a higher density (brighter appearance) than bone (bo). (**B**) 2D transverse reconstruction with section thickness of 10 µm, showing bone ongrowth on the coating material, without an ingrowth or growth of bone structures through the layer of the coating. All coating fragments were oriented in one plane and were 60–100 µm thick. Additionally, some small and globular fragments of the HA coating (size <20 µm) are identifiable.

**Figure 5 materials-13-04713-f005:**
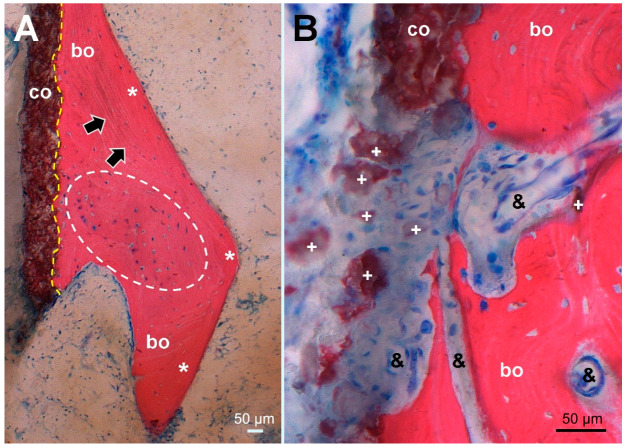
Histological picture (stained Giemsa–Eosin) of tissue sampled at the site corresponding to the dense line (arrowheads) in Figure 1C, maintaining the same orientation. (**A**) Overview of the HA coating (co, stained brown) showing firm bonding (dashed yellow line) to ongrown bone (bo, stained red) consisting of mature lamellar bone (*, black arrows marking lamellar orientation) surrounding newly formed woven bone (encircled by a white dashed line). In nearly all osteocyte lacunae, blue stained osteocyte nuclei indicate bone viability. (**B**) higher magnification of an HA coating defect with dispersed HA fragments (+) surrounded by fibrous scar tissue encompassing proliferated capillaries (&) and extending into resorption lacunae of remodeled woven bone without cellular inflammatory reaction.

**Figure 6 materials-13-04713-f006:**
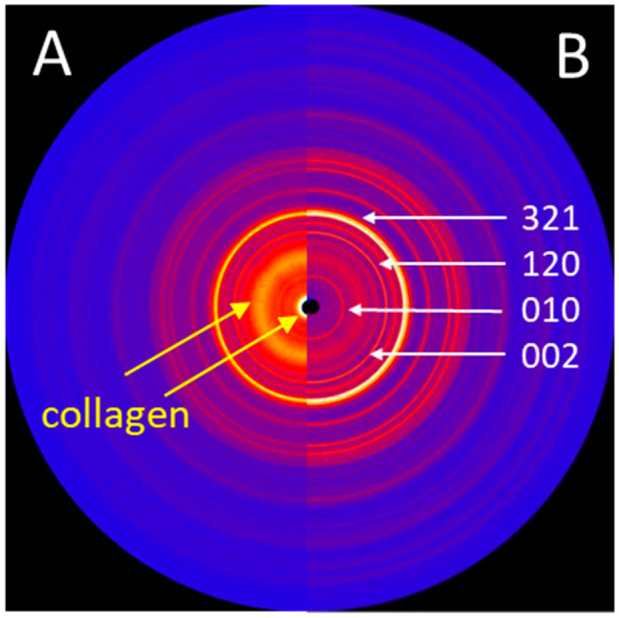
(**A**) 2D X-ray diffraction image (IPDS-II, Mo-radiation) of the trabecular bone structures, showing structural features of both HA and collagen. (**B**) 2D X-ray diffraction image of the HA powder obtained from the surface of a new stem, showing sharp diffraction peaks. The main diffraction peaks (hkl) of the HA crystallographic phase are indicated.

**Figure 7 materials-13-04713-f007:**
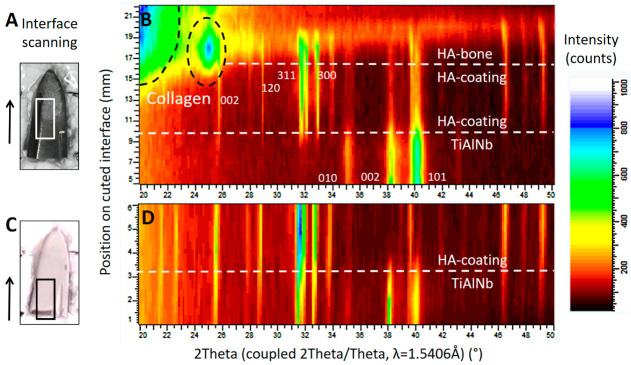
X-ray diffraction studies on the implant surfaces, in (**B**) the results corresponding to the area marked by a white rectangle from the tip of the retrieved stem in (**A**), and in (**D**) the results corresponding to the area marked by a black rectangle from the tip of a new, unused stem in (**C**). The scan was performed in the area of interest, following the direction marked by the arrow. The corresponding position may be read from the *y*-axis in (**B**) and (**D**). Both diffraction patterns (**B**) and (**D**) show the successive change from the TiAlNb of the stems’ metal alloy to the HA phase of the coating. The scan progresses horizontally on a slightly oblique cut. In (**B**), structures from the ongrown bone can be seen on top of the HA coating. (---white) shows the phase change related to the position on the sample, and dashed black lines show the phase information retrieved from the organic material such as collagen. The main diffraction peaks of the TiAlNb and HA crystallographic phases (hkl) are marked by the corresponding hkl values.

**Figure 8 materials-13-04713-f008:**
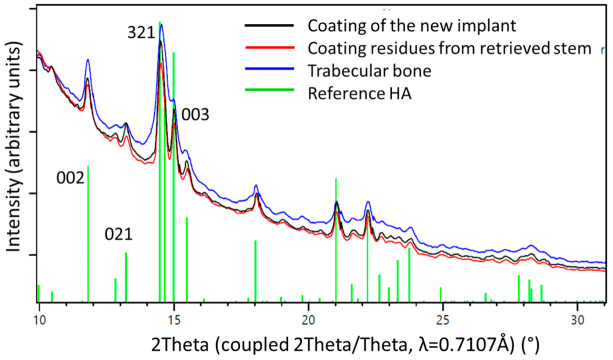
1D diffraction patterns obtained by a 360° integration of the 2D diffraction patterns (Figure 6). The indexing of the HA phase is shown in green. The peaks of the coating material from a pristine stem (black line) are higher and sharper than those from the materials residue recovered from the retrieved stem (red line) where the HA already was altered by biodegradation. The blue line marks results from bone trabeculae chipped of the retrieved stem. The observed peak broadening on the trabecular bone sample could be related to nano-sized crystallites and/or better nano-ordering due to strain probably formed during the growth process. Note the higher background signal from the bone sample, due to amorphous materials and proteins present within the tissue.

**Figure 9 materials-13-04713-f009:**
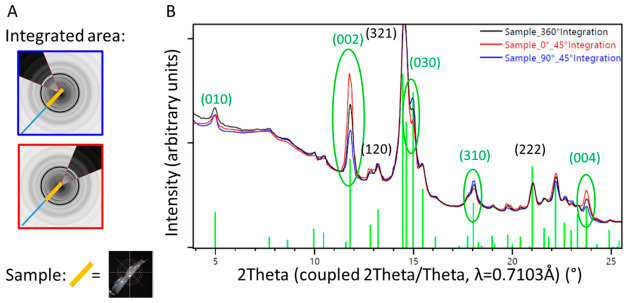
1D diffraction pattern of a sample of trabecular bone from the surface of the retrieved stem. Diffraction patterns from specific sectors (blue and red lines, corresponding to the orientation of the detector illustrated on the left) are compared with results from the 360° integration (black line). The orientation of the HA within the ongrown bone is directional, which can be visualized by different intensity profiles in dependence of the direction of investigation, differing from the random orientation observed in the 360° integration (black line). (**A**) Illustration of the partial integration of the 2D X-ray diffraction profile, blue perpendicular to the sample orientation, and red parallel to the sample orientation. (**B**) 1D diffraction profile superposed for the 360° (black) and 45° integration of the sectors illustrated in (**A**).

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
