# Peer review of "New Insights into Osteointegration and Delamination from a Multidisciplinary Investigation of a Failed Hydroxyapatite-Coated Hip Joint Replacement"

_materials, 2020, doi:10.3390/ma13214713_

Round 1
Reviewer 1 Report
The manuscript "New insights into osteointegration and delamination from a multidisciplinary investigation of a failed hydroxyapatite-coated hip joint replacement" report a case study on the used hydroxyapatite coatings that are very popular in uncemented total hip arthroplasty. It's a topical issue that can be of high interest to many readers. The manuscript is almost well written, some additional results being required. The manuscript can be considered for publication, after minor revision, according to the following comments:
- Page 5, Fig.3: What the magnification was set to obtain the image? What instrumentation was used for?
- Page 6, Fig.4: What the measure unit are in the presented images. It is recommended the unit measure to be written on the image.
- Page 8, Fig.6: the bright circles in 2D X-ray diffraction image B are observed to be slightly shifted. This means that the unit cell parameter and interplanar distance are of different values. May the authors explain/comment this aspect in the manuscript?
- The main diffraction peaks (hkl) of the hydroxyapatite crystallographic phase indicated in Fig.6 have to be found as well in the Fig.8, where the authors represented diffraction patterns of indexed HA phase.
- On page 10, the authors affirm that "Preferred orientation was found for the (001) crystallographic direction (see the change in the intensities of the reflections (002) and (004) in Fig.9..." being related directly to directional or preferential orientation of HA. As the intensities of some reflections in HA material change, may the authors calculate the particle/crystallite size using Scherrer equation? And correlate the crystallite size with the particle size determined from TEM, for example. In this case, are the particle, crystallite sizes the same as the natural bone?
- As well, what about the porosity of the newly formed HA?
Reviewer 2 Report
The paper reviewed represents a high quality investigation of a clinical case, supported by the results of analyses using several multidisciplinary analytical techniques. I think the paper could be published after minor revision of couple moments mentioned below.
- lines 252-257, 335-342: I would argue with these phrases. The sample has very anisotropic shape, which means that absorption of x-rays in various sectors is different, which in turn would result in intensities variations. To confirm these statements, you should use nearly isometric piece of your sample (ideally, but not necessary, spherical).
- In addition, I have some doubts on the absence of rotation during this experiment: It’s hard to believe that such an even 2D diffraction pattern could be obtained without rotation of the sample. May be it was not a Gandolfi mode, but it looks like some rotation (may be around its elongation axis - phi?) took place.
- line 233: wider or diffuse instead of larger.
- Of course, it would be very interesting to compare the results of osteointegration processes of implants with and without HA coatings after various time intervals, and of course it is not within the aim of the current work. But I, however, would address a question – how the disordered bone metabolism (in case of osteoporosis) affects the dissolution of HA from coating comparing to the “normal” cases?
Reviewer 3 Report
Materials Review – 935758
Within the manuscript a variety of techniques are applied in order to characterise the implant surface of a failed HA coated femoral stem. These techniques include photography, optical microscopy, micro-CT and X-ray diffraction. The manuscript considers only a single failed stem which the authors acknowledge was mal-aligned, so as a consequence the results within the manuscript cannot be considered representative of failure mechanisms of uncemented HA coated femoral stems in general. However this in itself does not mean the manuscript cannot be considered of scientific value or worthy of publication, I do however have some concerns with the data presented.
The first technique considered was ‘ the proper photodocumentation of the retrieval’ (as described within the discussion). However the paper includes only one photograph of the retrieved stem; could a macro lens have been used to get greater detail and multiple images, could the tissue fragments analysed also have been imaged.
Optical microscopy was then considered, but only a single micrograph was included, this time a patch of on-grown bone which your analysis would suggest is atypical of the surface structure. The manuscript could benefit from at the very least additional optical micrographs from other areas of the stem and ideally scanning electron microscopy images that would allow for better depth of field analysis and also the potential of chemical analysis through EDX. I appreciate SEM is not a technique that would typically be found within a clinical setting, but neither is micro-CT or XRD hence it does not feel unreasonable in suggesting it’s application.
Much more could also have been done with the micro-computed tomography. Could more than one fragment have been analysed particularly when it came to analysis of coating thickness. What evidence is there that the coating delaminated entirely from the femoral stem? What conclusive evidence is there to indicate the dense material is coating and not cortical type bone growth?
X-ray diffraction analysis is however systematic and rigorous.
Some specific points:
Introduction
Line 48 – ‘Even if questionable HA coatings are believed to accentuate osteointegration’ – First part of this statement must be substantiated if it is to be included, what evidence suggests it does not?
What is the significance of analysis of a single failed implant? Some acknowledgement of the limitations even within the introduction.
Method
Line 64 – what is the significance of the patient being deaf – does this risk being an identifying feature?
Methodology development but again can this be based solely upon a single sample?
Influence of processing on histological analysis?
Thorough X-ray diffraction analysis
Results
Macroscopic examination – was optical microscopy not carried out of areas of HA residue or polished stem – information from the macroscopic photograph is lacking.
Was bone apposition only observed on HA coating? CT from more areas of the explant tissue would have been beneficial. Is one fragment representative?
Fibrous scar tissue but no inflammatory response?
Ongrown bone on HA coating – should this come as a surprise?
Bone had grown onto most of the surface – no surprise that HA coating not generally visible
HA dissolved between bone trabeculae- where is evidence of this
Minimal evidence of fragmentation described – Was delamination proven?
Discussion
Bone had grown onto most of the surface – is this substantiated by the results included within the manuscript?
Line 285 - An osteoporotic bone in combination with an undersized stem, due to a subtle varus malalignment at the primary operation (Fig. 1), may be seen as contributing factor to the fracture – no evidence is provided of osteoporotic nature of the bone.
Round 2
Reviewer 3 Report
Whilst I appreciate the level of response to my comments on the manuscript, I must respectively stand by the comments raised. I simply do not feel this analysis of a single mal-aligned femoral stem is of sufficient clinical significance without further analysis
Author Response
The proposed manuscript remains a report from a single case, indeed. Even if full understanding of the failure mechanism required advanced analysis not available in clinical routine. Nevertheless, single case analysis remains an important tool to identify new or uncommon failure mechanisms. Many of the elements identified and described are not commonly known in orthopaedics and arthroplasty. As discussed, some of the elements had not been recognized in previous publications, even if identifiable on the illustrations provided. We remind that this case represented an exceptionally rare occasion to obtain appropriate samples, as discussed in the first paragraph of the Discussion. Also, we would like to underscore that this study provides new insights into osteointegration of HA coatings, and that the techniques described had not been applied for this indication so far and to the best of our knowledge. Additional results from further analysis would risk to overlengthen the manuscript, and the proposed SEM is not expected to provide relevant supplementary insights.
As we have difficulties identifying any new points or questions, our reply to the first review by Reviewer 3 is attached in full below. As Reviewer 1 and 2 obviously could be answered to satisfaction, we will not go into details of the points they raised. In order to better answer the points raised by Reviewer 3, following supplementary modifications have been made to the manuscript:
- A sentence mentioning that light reflections hindered obtaining microphotographs of the polished areas of the stem was added to section 3.1 of Results.
- Legend of Fig. 3: A sentence was added to mention that the same pattern was observable wherever bone had ongrown. Demultiplying pictures would not provide any new insights. One large picture is better in our opinion, as it allows better identification of small details.
- Legend of Fig. 4: A sentence was added, mentioning that identical patterns could be identified on all fragments. Here too, we believe one picture allows better identification of details, which would get lost when multiplying the images.
- Minor grammatical changes have been made on line 344 and line 360.
- The sentence in line 369 was added with a comment to underscore the relevance of single-case failure analysis.
Such a publication may nevertheless help surgeons identify relevant issues in other cases. This is the essential starting point required for sampling and retrieval analysis in further studies. The technique described should help accelerate analysis as the techniques used are described in detail. While this manuscript remains a single case analysis, we strongly believe it provides clinically relevant information and should serve as a useful reference for future studies.